# Reward prediction error does not explain movement selectivity in DMS-projecting dopamine neurons

Rachel S Lee, Marcelo G Mattar, Nathan F Parker, Ilana B Witten*, Nathaniel D Daw*

Department of Psychology, Princeton Neuroscience Institute, Princeton University, New Jersey, United States

**Abstract** Although midbrain dopamine (DA) neurons have been thought to primarily encode reward prediction error (RPE), recent studies have also found movement-related DAergic signals. For example, we recently reported that DA neurons in mice projecting to dorsomedial striatum are modulated by choices contralateral to the recording side. Here, we introduce, and ultimately reject, a candidate resolution for the puzzling RPE vs movement dichotomy, by showing how seemingly movement-related activity might be explained by an action-specific RPE. By considering both choice and RPE on a trial-by-trial basis, we find that DA signals are modulated by contralateral choice in a manner that is distinct from RPE, implying that choice encoding is better explained by movement direction. This fundamental separation between RPE and movement encoding may help shed light on the diversity of functions and dysfunctions of the DA system.
DOI: https://doi.org/10.7554/eLife.42992.001

*For correspondence:
iwitten@princeton.edu (IBW);
ndaw@princeton.edu (NDD)

Competing interests: The authors declare that no competing interests exist.

## Introduction

A central feature of dopamine (DA) is its association with two apparently distinct functions: reward and movement (*Niv et al., 2007*; *Berke, 2018*). Although manipulation of DA produces gross effects on movement initiation and invigoration, physiological recordings of DA neurons have historically shown few neural correlates of motor events (*Wise, 2004*; *Schultz et al., 1997*). Instead, classic studies reported responses to rewards and reward-predicting cues, with a pattern suggesting that DA neurons carry a 'reward prediction error' (RPE) — the difference between expected reward and observed reward — for learning to anticipate rewards (*Schultz et al., 1997*; *Barto, 1995*; *Cohen et al., 2012*; *Coddington and Dudman, 2018*; *Soares et al., 2016*; *Hart et al., 2014*). In this classic framework, rather than explicitly encoding movement, DA neurons influence movements indirectly by determining which movements are learned and/or determining the general motivation to engage in a movement (*Niv et al., 2007*; *Collins and Frank, 2014*; *Berke, 2018*).

Complicating this classic view, however, several recent studies have suggested that subpopulations of DA neurons may have a more direct role in encoding movement. For example, we recently reported that whereas DA neurons projecting to ventral striatum showed classic RPE signals, a subset of midbrain DA neurons that project to the dorsomedial striatum (DMS) were selective for a mouse's choice of action (*Parker et al., 2016*). In particular, they responded more strongly during contralateral (versus ipsilateral) choices as mice performed a probabilistic learning task (*Parker et al., 2016*). In addition, there have been several other recent studies that reported phasic changes in DA activity at the onset of spontaneous movements (*Dodson et al., 2016*; *Howe and Dombeck, 2016*; *Howe et al., 2013*; *da Silva et al., 2018*; *Barter et al., 2015*; *Syed et al., 2016*). Moreover, other studies have shown that DA neurons may have other forms of apparently non-RPE

signals, such as signals related to novel or aversive stimuli (*Menegas et al., 2017*; *Horvitz, 2000*; *Ungless et al., 2004*; *Matsumoto and Hikosaka, 2009*; *Lammel et al., 2011*).

These recent observations of movement selectivity leave open an important question: can the putatively movement-related signals be reconciled with Reinforcement Learning (RL) models describing the classic RPE signal? For instance, while it seems plausible that movement-related DA signals could influence movement via directly modulating striatal medium spiny neurons (*DeLong, 1990*), these signals are accompanied in the same recordings by RPEs which are thought to drive cortico-striatal plasticity (*Reynolds et al., 2001*). It is unclear how these two qualitatively different messages could be teased apart by the recipient neurons. Here we introduce and test one possible answer to this question, which we argue is left open by *Parker et al. (2016)* results and also by other reports of movement-related DA activity: that these movement-related signals actually also reflect RPEs, but for reward predictions tied to particular movement directions. Specifically, computational models like advantage learning (*Baird, 1994*) and actor-critic (*Barto et al., 1983*) learn separate predictions about the overall value of situations or stimuli and about the value of specific actions. It has long been suggested these two calculations might be localized to ventral vs dorsal striatum respectively (*Montague et al., 1996*; *O'Doherty et al., 2004*; *Takahashi et al., 2008*). Furthermore, a human neuroimaging experiment reported evidence of distinct prediction errors for right and left movements in the corresponding contralateral striatum (*Gershman et al., 2009*).

This leads to the specific hypothesis that putative movement-related signals in DMS-projecting DA neurons might actually reflect an RPE related to the predicted value of contralateral choices. If so, this would unify two seemingly distinct messages observed in DA activity. Importantly, a choice-specific RPE could explain choice-related correlates observed prior to the time of reward. This is because temporal difference RPEs do not just signal error when a reward is received, they also have a phasic anticipatory component triggered by predictive cues indicating the availability and timing of future reward, such as (in choice tasks) the presentation of levers or choice targets (*Montague et al., 1996*; *Morris et al., 2006*; *Roesch et al., 2007*). This anticipatory prediction error is proportional to the value of the future expected reward following a given choice — indeed, we henceforth refer to this component of the RPE as a 'value' signal, which tracks the reward expected for a choice. Crucially, a choice-specific value signal can masquerade as a choice signal because, by definition, action and value are closely related to each other: animals are more likely to choose actions that they predict have high value. In this case, a value signal (RPE) for the contralateral choice will tend to be larger when that action is chosen than when it is not (*Samuelson, 1938*). Altogether, given the fundamental correlation between actions and predicted value, a careful examination of the neural representation of both quantities and a clear understanding of if and how they can be differentiated is required to determine whether or not movement direction signals can be better explained as value-related.

Thus, we examined whether DA signals in DMS-projecting DA neurons are better understood as a contralateral movement signal or as a contralateral RPE. To tease apart these two possibilities, we measured neural correlates of value and lateralized movement in our DA recordings from mice performing a probabilistic learning task. Since value predictions are subjective, we estimated value in two ways: (1) by using reward on the previous trial as a simple, theory-neutral proxy, and (2) by fitting the behavioral data with a more elaborate trial-by-trial Q-learning model. We compared the observed DA modulations to predictions based on modulation either by movement direction and/or the expected value (anticipatory RPE) of contralateral or chosen actions.

Ultimately, our results show that DMS-projecting DA neurons' signals are indeed modulated by value (RPE), but, crucially, this modulation reflected the value of the chosen action rather than the contralateral one. Thus, the value aspects of the signals (which were not lateralized) could not explain the contralateral choice selectivity in these neurons, implying that this choice-dependent modulation indeed reflects modulation by contralateral movements and not value.

## Results

### Task, behavior and DA recordings

Mice were trained on a probabilistic reversal learning task as reported previously (*Parker et al., 2016*). Each trial began with an illumination in the nose port, which cued the mouse to initiate a

nose poke (*Figure 1a*). After a 0–1 second delay, two levers appeared on both sides of the nose port. Each lever led to reward either with high probability (70%) or low probability (10%), with the identity of the high probability lever swapping after a block of variable length (see Materials and methods for more details, *Figure 1b*). After another 0–1 second delay, the mouse either received a sucrose reward and an accompanying auditory stimulus (positive conditioned stimulus, or CS+), or no reward and a different auditory stimulus (negative conditioned stimulus, or CS-).

Given that block transitions were not signaled to the mouse, mice gradually learned to prefer the lever with the higher chance of reward after each transition. To capture this learning, we fit their choices using a standard trial-by-trial Q-learning model that predicted the probability of the animal's choice at each trial of the task (*Figure 1c*, *Table 1*). In the model, these choices were driven by a pair of decision variables (known as Q-values) putatively reflecting the animal's valuation of each option.

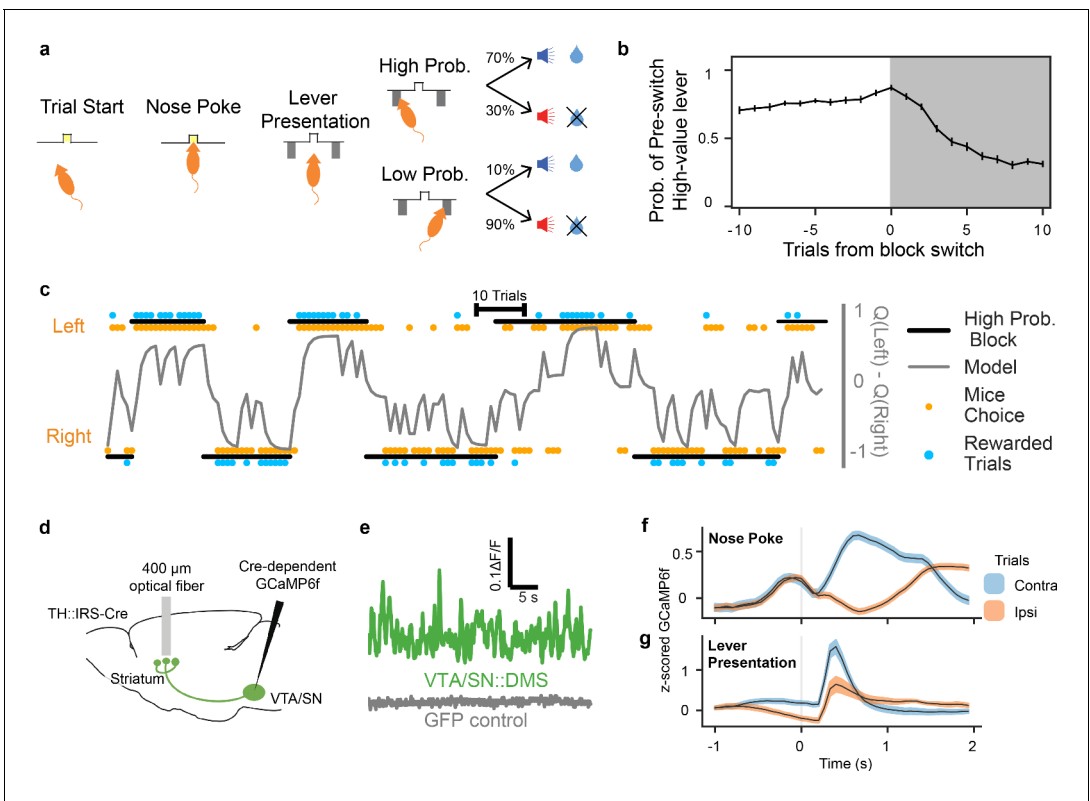

**Figure 1.** Mice performed a probabilistic reversal learning task during GCaMP6f recordings from VTA/SN::DMS terminals or cell bodies. (**a**) Schematic of a mouse performing the task. The illumination of the central nosepoke signaled the start of the trial, allowing the mouse to enter the nose port. After a 0–1 second jitter delay, two levers were presented to the mouse, one of which result in a reward with high probability (70%) and the other with a low probability (10%). The levers swapped probabilities on a pseudorandom schedule, unsignaled to the mouse. (**b**) The averaged probability of choosing the lever with high value before the switch, 10 trials before and after the block switch, when the identity of the high value lever reversed. Error bars indicate ±1 standard error (n = 19 recording sites). (**c**) We fit behavior with a trial-by-trial Q learning mixed effect model. Example trace of 150 trials of a mouse's behavior compared to the model's results. Black bars above and below the plot indicate which lever had the high probability for reward; Orange dots indicate the mouse's actual choice; Blue dots indicate whether or not mouse was rewarded; Grey line indicates the difference in the model's Q values for contralateral and ipsilateral choices. (**d**) Surgical schematic for recording with optical fibers from the GCaMP6f terminals originating from VTA/SN. (**e**) Example recording from VTA/SN::DMS terminals in a mouse expressing GCaMP6f (top) or GFP (bottom). (**f, g**) Previous work has reported contralateral choice selectivity in VTA/SN::DMS terminals (*Parker et al., 2016*) when the signals are time-locked to nose poke (**f**) and lever presentation (**g**). 'Contra' and 'Ipsi' refer to the location of the lever relative to the side of the recording. Colored fringes represent ±1 standard error (n=12 recording sites).

DOI: https://doi.org/10.7554/eLife.42992.002

The following figure supplement is available for figure 1:

**Figure supplement 1.** Recording from VTA/SN::DMS cell bodies (n = 7 recording sites).

DOI: https://doi.org/10.7554/eLife.42992.003

**Table 1.** Fitted Parameters for Q-learning model from PyStan.

25th, 50th, and 75th percentile of the alpha, beta, and stay parameters of the Q-learning mixed effect model. These are the the group-level parameters that reflect the distribution of the subject-level parameters.

|  | 25th percentile | 50th percentile (median) | 75th percentile |
|---|---|---|---|
| Alpha (learning rate) | 0.581607 | 0.611693 | 0.639946 |
| Beta (inverse temperature) | 0.926501 | 0.990275 | 1.058405 |
| Stay | 0.883670 | 0.945385 | 1.008465 |

DOI: https://doi.org/10.7554/eLife.42992.004
The following source data is available for Table 1:
Source data 1. Mixed effect Q-learning model parameters.
Parameters from the mixed effect Q-learning model, including group-level and individual-level parameters, and the mean and range of data across samples from the model. See 'Q Learning Mixed Effect Model' in the Materials and methods section for more details.
DOI: https://doi.org/10.7554/eLife.42992.005

As mice performed this task, we recorded activity from either the terminals or cell bodies of DA neurons that project to DMS (VTA/SN::DMS) using fiber photometry to measure the fluorescence of the calcium indicator GCaMP6f (*Figure 1d,e*; *Figure 1—figure supplement 1a,b*). As previously reported, this revealed elevated activity during contralateral choice trials relative to ipsilateral choice trials, particularly in relation to the nose poke and lever presentation events (*Figure 1f,g*; *Figure 1— figure supplement 1c*) (*Parker et al., 2016*).

## Predictions of contralateral and chosen value models

In order to examine how value-related activity might (or might not) explain seemingly movement-related activity, we introduced two hypothetical frames of reference by which the DMS DA neurons' activity may be modulated by predicted value during trial events prior to the outcome: the DA signals could be modulated by the value of the contralateral option (relative to ipsilateral; *Figure 2a*) or by the value of the chosen option (relative to unchosen; *Figure 2b*). Note that both of these modulations could be understood as the anticipatory component (occasioned at lever presentation) of a temporal difference RPE, with respect to the respective action's value.

The first possibility is modulation by the value of the contralateral (relative to ipsilateral) action (*Figure 2a*; such signals have been reported in human neuroimaging, [*Gershman et al., 2009*, *Palminteri et al., 2009*] but not previously, to our knowledge examined in DA recordings in animals). The motivation for this hypothesis is that, if neurons in DMS participate in contralateral movements, such a side-specific error signal would be appropriate for teaching them when those movements are valuable. In this case, the relative value of the contralateral (versus ipsilateral) choice modulates DA signals, regardless of whether the choice is contralateral or ipsilateral. Thus, when the DA signals are broken down with respect to both the action chosen and its value, the direction of value modulation would depend on the choice: signals are highest for contralateral choices when these are relatively most valuable, but lowest for ipsilateral choices when *they* are most valuable (because in this case, contralateral choices will be relatively less valuable). Assuming mice tend to choose the option they expect to deliver more reward, such signals would be larger, on average, during contralateral choices than ipsilateral ones (*Figure 2a*), which could in theory explain the contralateral choice selectivity that we observed (*Figure 1f,g*).

The second possibility is that value modulation is relative to the chosen (versus unchosen) option (*Figure 2b*). This corresponds to the standard type of 'critic' RPE most often invoked in models of DA: that is, RPE with respect to the overall value of the current state or situation (where that state reflects any choices previously made), and not specialized to a particular class of action. Indeed, human neuroimaging studies have primarily reported correlates of the value of the chosen option in DA target areas (*Daw et al., 2006*; *Boorman et al., 2009*; *Li and Daw, 2011*), and this also has been observed in primate DA neurons (*Morris et al., 2006*).

If DMS-projecting DA neurons indeed display chosen value modulation (*Figure 2b*), rather than contralateral value modulation, the value modulation for both contralateral and ipsilateral choices

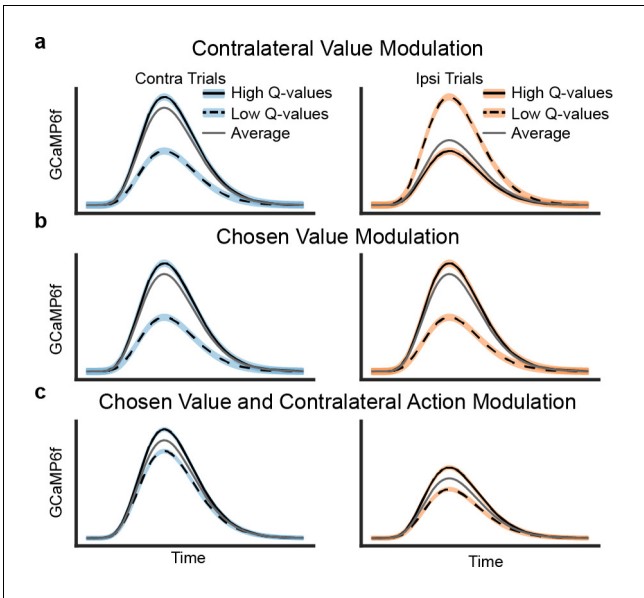

**Figure 2.** Schematics of three possible types of value modulation at lever presentation. Trials here are divided based on the difference in Q values for chosen and unchosen action. (a) Contralateral value modulation postulates that the signals are selective for the *value* of the contralateral action (relative to ipsilateral value) instead of the action chosen. This means that the direction of value modulation should be flipped for contralateral versus ipsilateral choices. Since mice would more often choose an option when its value is higher, the average GCaMP6f signals would be higher for contralateral than ipsilateral choices. (b) Alternatively, the signals may be modulated by the value of the chosen action, resulting in similar value modulation for contralateral and ipsilateral choices. This type of value modulation will not in itself produce contralateral selectivity seen in previous results. (c) However, if the signals were modulated by the chosen value and the contralateral choice, the averaged GCaMP6f would exhibit the previously seen contralateral selectivity.

DOI: https://doi.org/10.7554/eLife.42992.006

would be similar. In this case, value modulation could not in itself account for the neurons' elevated activity during contralateral trials, which we have previously observed (*Figure 1f,g*). Therefore, to account for contralateral choice preference, one would have to assume DA neurons are also selective for the contralateral action itself (unrelated to their value modulation; *Figure 2c*).

## DA in dorsomedial striatum is modulated by chosen value, not contralateral value

Next, we determined which type of value modulation better captured the signal in DA neurons that project to DMS by comparing the GCaMP6f signal in these neurons for high and low value trials. We focused on the lever presentation since this event displayed a clear contralateral preference (*Figure 1g*). As a simple and objective proxy for the value of each action (i.e., the component of the RPE at lever presentation for each action), we compared signals when the animal was rewarded (high value), or not (low value) on the previous trial. (To simplify the interpretation of this comparison, we only included trials in which the mice made the same choice as the preceding trial, which accounted for 76.6% of the trials.) The traces (*Figure 3a*) indicated that the VTA/SN::DMS terminals were modulated by the previous trial's reward. The value-related signals reflected chosen value — responding more when the previous choice was rewarded, whether contralateral or ipsilateral — and therefore did not explain the movement-related effect. This indicated that the DMS-projecting DA neurons represented both chosen value and movement direction (similar to *Figure 2c*). The effect of contralateral action modulation was also visible in individual, non-z-scored data in both VTA/SN:: DMS terminals (*Figure 3—figure supplement 1*) and VTA/SN::DMS cell-bodies (*Figure 3—figure supplement 2*).

We repeated this analysis using trial-by-trial Q values extracted from the model, which we reasoned should reflect a finer grained (though more assumption-laden) estimate of the action's value.

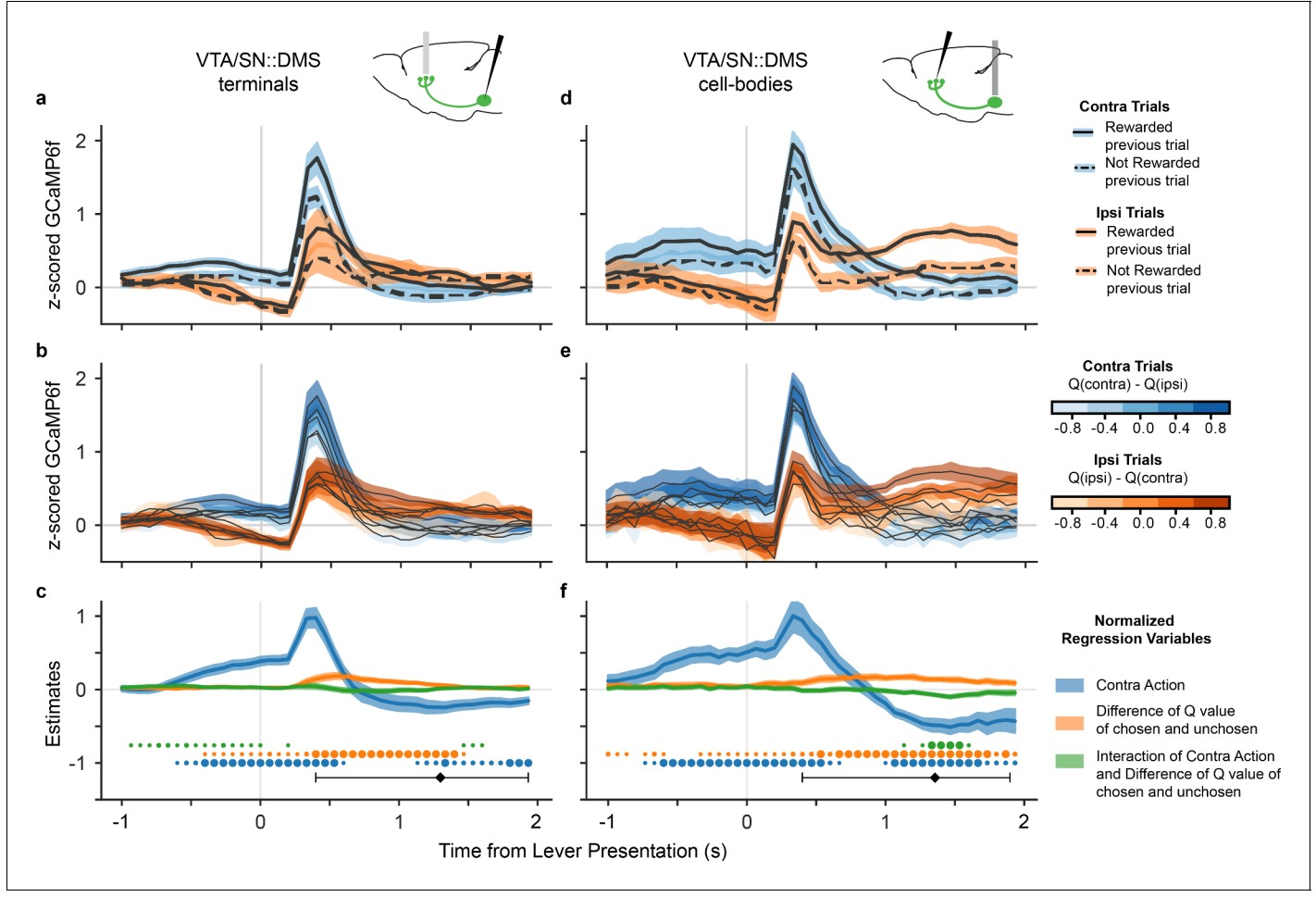

**Figure 3.** DA neurons that project to DMS were modulated by both chosen value and movement direction. (**a**) GCaMP6f signal time-locked to lever presentation for contralateral trials (blue) and ipsilateral trials (orange), as well as rewarded (solid) and non-rewarded previous trial (dotted) from VTA/SN::DMS terminals. Colored fringes represent ±1 standard error from activity averaged across recording sites (n = 12). (**b**) GCaMP6f signal for contralateral trials (blue) and ipsilateral trials (orange), further binned by the difference in Q values for chosen and unchosen action. Colored fringes represent ±1 standard error from activity averaged across recording sites (n = 12). (**c**) Mixed effect model regression on each datapoint from 3 seconds of GCaMP6f traces. Explanatory variables include the action of the mice (blue), the difference in Q values for chosen and unchosen actions (orange), their interaction (green), and an intercept. Colored fringes represent ±1 standard error from estimates (n = 12 recording sites). Black diamond represents the average latency for mice pressing the lever, with the error bars showing the spread of 80% of the latency values. Dots at bottom mark timepoints when the corresponding effect is significantly different from zero at p<0.05 (small dot), p<0.01 (medium dot), p<0.001 (large dot). P values were corrected with Benjamini Hochberg procedure. (**d-f**) Same as (**a-e**), except with signals from VTA/SN::DMS cell bodies averaged across recording sites (n = 7) instead of terminals.

DOI: https://doi.org/10.7554/eLife.42992.007

The following figure supplements are available for figure 3:

**Figure supplement 1.** Four Examples of non-Z-scored Individual Sessions of Photometry Data from VTA/SN::DMS Terminals.
DOI: https://doi.org/10.7554/eLife.42992.008

**Figure supplement 2.** Four Examples of non-Z-scored Individual Sessions of Photometry Data from VTA/SN::DMS Cell-Bodies.
DOI: https://doi.org/10.7554/eLife.42992.009

**Figure supplement 3.** Mixed effect model regression on GCaMP6f traces of VTA/SN::DMS terminals (n = 12 recording sites) using the difference in Q values for contralateral and ipsilateral choices.
DOI: https://doi.org/10.7554/eLife.42992.010

**Figure supplement 4.** Analysis of DA signals time-locked to nose poke.
DOI: https://doi.org/10.7554/eLife.42992.011

**Figure supplement 5.** Kernels for each significant behavioral event from the multiple event kernel analysis.
DOI: https://doi.org/10.7554/eLife.42992.012

*Figure 3 continued on next page*

*Figure 3 continued*

**Figure supplement 6.** Averaged GCaMP6f signals of left and right hemispheres recordings from VTA/SN::DMS cell-bodies data (n = 4 mice, 7 recording sites).

DOI: https://doi.org/10.7554/eLife.42992.013

**Figure supplement 7.** Mixed effect model regression with latency as nuisance covariate.

DOI: https://doi.org/10.7554/eLife.42992.014

(For this analysis, we were able to include both stay and switch trials.) Binning trials by chosen (minus unchosen) value, a similar movement effect and value gradient emerged as we have seen with the previous trial outcome analysis (*Figure 3b*). Trials with higher Q values had larger GCaMP6f signals, regardless which side was chosen, again suggesting that VTA/SN::DMS terminals were modulated by the expected value of the chosen (not contralateral) action, in addition to being modulated by contralateral movement.

To quantify these effects statistically, we used a linear mixed effects regression at each time point of the time-locked GCaMP6f. The explanatory variables included the action chosen (contra or ipsi), the differential Q values (oriented in the reference frame suggested by the data, chosen minus unchosen), the value by action interaction, and an intercept (*Figure 3c*). The results verify significant effects for both movement direction and action value; that is, although a significant value effect is seen, it does not explain away the movement effect. Furthermore, the appearance of a consistent chosen value effect across both ipsilateral and contralateral choices is reflected in a significant value effect and no significant interaction during the period when action and value coding are most prominent (0.25–1 seconds after lever presentation), as would have been predicted by the contralateral value model. (There is a small interaction between the variables earlier in the trial, before 0.25 seconds, reflecting small differences in the magnitude of value modulation on contralateral versus ipsilateral trials.) Conversely, when the regression is re-estimated in terms of contralateral value rather than chosen value, a sustained, significant interaction does emerge, providing formal statistical support for the chosen value model; see *Figure 3—figure supplement 3*.

We performed the same value modulation analyses on the cell bodies, rather than terminals, of VTA/SN::DMS neurons (*Figure 3d–f*). This was motivated by the possibility that there may be changes in neural coding between DA cell bodies and terminals due to direct activation of DA terminals. In this case, we found very similar modulation by both chosen value and contralateral movement in both recording locations.

To verify the robustness of these findings, we conducted further followup analyses. In one set of analyses, we investigated to what extent the DA signals might be tied to particular events other than the lever presentation. First, we repeated our analyses on DA signals time-locked to nose poke event (*Figure 3—figure supplement 4*) and found the same basic pattern of effects. The effect was still clearest close to the average lever presentation latency, suggesting that the modulation of DA signals is more closely related to lever presentation. To more directly verify that our conclusions are independent of the specific choice event alignment, we fit a linear regression model with kernels capturing the contribution of three different events (Nose Poke, Lever Presentation, and Lever Press) simultaneously (*Figure 3—figure supplement 5*). The results of this multiple event regression were consistent with the simpler single-event regression in *Figure 3a,d*.

Next, we examined a few other factors that might have affected movement-specific activity. Taking advantage of the fact that the VTA/SN::DMS cell-bodies data had recordings from both hemispheres in three animals, we directly compared signals across hemispheres in individual mice and observed that the side-specific effects reversed within animal (*Figure 3—figure supplement 6*). This speaks against the possibility that they might reflect animal-specific idiosyncrasies such as side biases. Finally, we considered whether the contralateral action modulation might in part reflect movement vigor rather than action value. We addressed this by repeating the analysis in *Figure 3c, f*, but including as an additional covariate the log lever-press latency as a measure of the action's vigor. For both VTA/SN::DMS terminals and cell-bodies data, the lever-press latency was not a strong predictor for GCaMP6f signals, and the effect of the original predictors largely remained the same (*Figure 3—figure supplement 7*).

## Direction of movement predicts DMS DA signals

An additional observation supported the interpretation that the contralateral choice selectivity in DMS-projecting DA neurons is related to the direction of movement and not the value of the choice. When the signals were time-locked to the lever press itself, there was a reversal of the signal selectivity between contralateral and ipsilateral trials, shortly after the lever press (*Figure 4*). Although body tracking is not available, this event coincided with a reversal in the animal's physical movement direction, from moving towards the lever from the central nosepoke before the lever press, to moving back to the central reward port after the lever press. In contrast, there is no reversal in value modulation at the time of lever press. The fact that the side-specific modulation (and not the value modulation) followed the mice's movement direction during the trial further indicated that movement direction explains the choice selectivity in these DA neurons, and resists explanation in terms of RPE-related signaling.

## Discussion

Recent reports of qualitatively distinct DA signals — movement and RPE-related — have revived perennial puzzles about how the system contributes to both movement and reward, and more specifically raise the question whether there might be a unified computational description of both components in the spirit of the classic RPE models (*Parker et al., 2016*; *Berke, 2018*; *Coddington and Dudman, 2018*; *Syed et al., 2016*). Here we introduce and test one possible route to such a unification: action-specific RPEs, which could explain seemingly action-selective signals as instead reflecting RPE related to the value of those actions. To investigate this possibility, we dissected movement direction and value selectivity in the signals of terminals and cell bodies of DMS-projecting DA neurons (*Figure 3*). Contrary to the hypothesis that lateralized movement-related activity might reflect a RPE for contralateral value, multiple lines of evidence clearly indicated that the neurons instead contain distinct movement- and value-related signals, tied to different frames of reference. We did observe value-related signals preceding and following the lever press, which we did not previously analyze in the DMS signal and which are consistent with the anticipatory component of a classic RPE signal (*Parker et al., 2016*). But because these were modulated by the value of the chosen action, not the contralateral one, they cannot explain the side-specific movement selectivity. The two signals

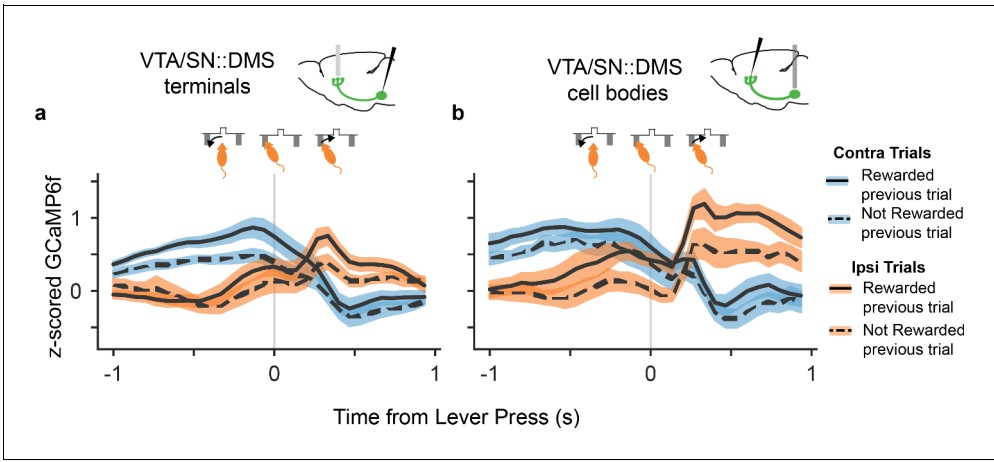

**Figure 4.** DA neurons that project to DMS reversed their choice selectivity after the lever press, around the time the mice reversed their movement direction. (a). GCaMP6f signal from VTA/SN::DMS terminals time-locked to the lever press, for contralateral choice trials (blue) and ipsilateral choice trials (orange), as well as rewarded (solid) and non-rewarded previous trial (dotted). The GCaMP6f traces for each choice cross shortly after the lever-press, corresponding to the change in the mice's head direction around the time of the lever press (shown schematically above the plot). Colored fringes represent ±1 standard error from activity averaged across recording sites (n = 12). (b) Same as (a), except with signals from VTA/SN::DMS cell bodies averaged across recording sites (n = 7) instead of terminals.

DOI: https://doi.org/10.7554/eLife.42992.015

also showed clearly distinct time courses; in particular, the side selectivity reversed polarity following the lever press, but value modulation did not.

Our hypothesis that apparently movement-related DA correlates might instead reflect action-specific RPEs (and our approach to test it by contrasting chosen vs. action-specific value) may also be relevant to other reports of DAergic movement selectivity. For example, Syed et al. recently reported that DA release in the nucleus accumbens (NAcc) was elevated during 'go', rather than 'no-go', responses, alongside classic RPE-related signals (*Syed et al., 2016*). This study in a question analogous to the one we raise about Parker's (*Parker et al., 2016*) DMS results: could NAcc DA instead reflect an RPE specific for 'go' actions? This possibility would be consistent with the structure's involvement in appetitive approach and invigoration (*Parkinson et al., 2002*), and might unify the RPE- and 'go'-related activity reported there via an action-specific RPE (argument analogous to *Figure 2a*). The analyses in the Syed et al. study did not formally compare chosen- vs. action-specific value, and much of the reward-related activity reported there appears consistent with either account (*Syed et al., 2016*). However, viewed from the perspective of our current work, the key question becomes whether the value-related DA signals on 'go' cues reverses for 'no-go' cues, as would be predicted for an action-specific RPE. There is at least a hint (albeit significant only at one timepoint in Syed et al.'s Supplemental Figure 9E) that it does not do so (*Syed et al., 2016*). This suggests that NAcc may also have parallel movement-specific and chosen value signals, which would be broadly confirmatory for our parallel conclusions about DMS-projecting DA neurons.

The RPE account of the DA signal has long held out hope for a unifying perspective on the system's dual roles in movement and reward by proposing that the system's reward-related signals ultimately affect movement indirectly, either by driving learning about movement direction preferences (*Montague et al., 1996*) or by modulating motivation to act (*Niv et al., 2007*). This RPE theory also accounts for multiple seemingly distinct components of the classic DA signal, including anticipatory and reward-related signals, and signals to novel neutral cues. However, the present analyses clearly show that side-specific signals in DMS resist explanation in terms of an extended RPE account, and may instead simply reflect planned or ongoing movements.

Specifically, our results are consistent with the longstanding suggestion that DA signals may be important for directly initiating movement. Such a signal may elicit or execute contralateral movements via differentially modulating the direct and indirect pathways out of the striatum (*Alexander and Crutcher, 1990*; *Collins and Frank, 2014*; *DeLong, 1990*). The relationship between unilateral DA activity and contralateral movements is also supported by causal manipulations. For instance, classic results demonstrate that unilateral 6-hydroxydopamine (6-OHDA) lesions increase ipsilateral rotations (*Costall et al., 1976*; *Ungerstedt and Arbuthnott, 1970*). Consistent with those results, a recent study reports that unilateral optogenetic excitation of midbrain DA neurons in mice led to contralateral rotations developed over the course of days (*Saunders et al., 2018*). Importantly, however, our own results are correlational, and we cannot rule out the possibility that the particular activity we study could be related to a range of functions other than movement execution, such as planning or monitoring. Another function that is difficult to distinguish from movement execution is the motivation to move. Although motivation is a broad concept and difficult to operationalize fully, our results address two aspects of it. First, one way to quantify the motivation to act is by the action's predicted value; thus, our main result is to rule out the possibility that neural activity is better accounted for by this motivational variable. We also show that lever press latency (arguably another proxy for motivation) does not explain the contralateralized DA signals (*Figure 3—figure supplement 7*).

Although the movement-related DA signal might be appropriate for execution, it is less clear how it might interact with the plasticity mechanisms hypothesized to be modulated by RPE aspects of the DA signal (*Frank et al., 2004*; *Steinberg et al., 2013*; *Reynolds and Wickens, 2002*). For instance, how would recipient synapses distinguish an RPE component of the signal (appropriate for surprise-modulated learning) from an overlapping component more relevant to movement elicitation (*Berke, 2018*)? We have ruled out the possibility that the activity is actually a single RPE for action value, but there may still be other sorts of plasticity that might be usefully driven by a purely movement-related signal. One possibility is that plasticity in the dorsal striatum itself follows different rules, which might require an action rather than a prediction error signal (*Saunders et al., 2018*; *Yttri and Dudman, 2016*) For instance, it has been suggested that some types of instrumental learning are correlational rather than error-driven (*Doeller et al., 2008*) and, more specifically, an early

model of instrumental learning (*Guthrie, 1935*) recently revived by (*Miller et al., 2019*) posits that stimulus-response habits are not learned from an action's rewarding consequences, as in RPE models, but instead by directly memorizing which actions the organism tends to select in a situation. Although habits are more often linked to adjacent dorsolateral striatum (*Yin et al., 2004*), a movement signal of the sort described here might be useful to drive this sort of learning. Investigating this suggestion will likely require new experiments centered around causal manipulations of the signal. Overall, our results point to the need for an extended computational account that incorporates the movement direction signals as well as the RPE ones.

Another striking aspect of the results is the co-occurrence of two distinct frames of reference in the signal. Lateralized movement selectivity tracks choices contralateral versus ipsilateral of the recorded hemisphere — appropriate for motor control — but the value component instead relates to the reward expected for the chosen, versus unchosen, action. This value modulation by the chosen action is suitable for a classic RPE for learning 'state' values (since overall value expectancy at any point in time is conditioned on the choices the animal has made; *Morris et al., 2006*), and also consistent with the bulk of BOLD signals in human neuroimaging, where value-related responding throughout DAergic targets tends to be organized on chosen-vs-unchosen lines (*Daw et al., 2006*; *Boorman et al., 2009*; *Li and Daw, 2011*; *O'Doherty, 2014*).

At the same time, there have been persistent suggestions that given the high dimensionality of an organism's action space, distinct action-specific error signals would be useful for learning about different actions (*Russell and Zimdars, 2003*; *Frank and Badre, 2012*; *Diuk et al., 2013*) or types of predictions (*Gardner et al., 2018*; *Lau et al., 2017*). Along these lines, there is evidence from BOLD neuroimaging for contralateral error and value signals in the human brain (*Gershman et al., 2009*; *Palminteri et al., 2009*). Here, we have shown how a similar decomposition might explain movement-related DA signals, and also clarified how this hypothesis can be definitively tested. Although the current study finds no evidence for such laterally decomposed RPEs in DMS, the decomposition of error signals remains an important possibility for future work aimed at understanding heterogeneity of DA signals, including other anomalous features like ramps (*Howe et al., 2013*; *Berke, 2018*; *Gershman, 2014*; *Hamid et al., 2016*; *Engelhard et al., 2018*). Recent studies, for instance, have shown that midbrain DA neurons may also encode a range of behavioral variables, such as the mice's position, their velocity, their view-angle, and the accuracy of their performance (*da Silva et al., 2018*; *Howe et al., 2013 Engelhard et al., 2018*). Our modeling provides a framework for understanding how these DA signals might be interpreted in different reference frames and how they might ultimately encode some form of RPEs with respect to different behavioral variables in the task.

Interestingly, our results were consistent across both recording sites with DMS-projecting DA neurons: the cell bodies and the terminals (*Figure 3d–f*, *Figure 4b*). This indicates that the movement selectivity is not introduced in DA neurons at the terminal level, for example via striatal cholinergic interneurons or glutamatergic inputs (*Kosillo et al., 2016*).

An important limitation of the study is the use of fiber photometry, which assesses bulk GCaMP6f signals at the recording site rather than resolving individual neurons. Thus it remains possible that individual neurons do not multiplex the two signals we observe, and that they are instead segregated between distinct populations. Future work should use higher resolution methods to examine these questions at the level of individual DA neurons. A related limitation of this study is the relatively coarse behavioral monitoring; notably, we infer that the reversal in selectivity seen in *Figure 4* reflects a change in movement direction, but head tracking would be required to verify this more directly. More generally, future work with finer instrumentation could usefully dissect signal components related to finer-grained movements, and examine how these are related to (or dissociated from) value signals.

## Materials and methods

### Mice and surgeries

This article reports new analysis on data originally reported by (*Parker et al., 2016*). We briefly summarize the methods from that study here. This article reports on data from 17 male mice expressing

Cre recombinase under the control of the tyrosine hydroxylase promoter ($Th^{\text{IRES-Cre}}$), from which GCaMP6f recordings were obtained from DA neurons via fiber photometry.

In the case of DA terminal recordings, Cre-dependent GCaMP6f virus (AAV5-CAG-Flex-GCamp6f-WPRE-SV40; UPenn virus core, 500nL, titer of $3.53 \times 10^{12}$ pp ml) was injected into the VTA/SNc, and fibers were placed in the DMS (M–L ± 1.5, A–P 0.74 and D–V −2.4 mm), with one recording area per mouse (n = 12 recording sites). The recording hemisphere was counterbalanced across mice. The mice were recorded bilaterally, with the second site in nucleus accumbens, which is not analyzed in this paper.

In the case of VTA/SN::DMS cell body recordings, Cre-dependent GCaMP6f virus (AAV5-CAG-Flex-GCamp6f-WPRE-SV40; UPenn virus core, 500nL, titer of $3.53 \times 10^{12}$ pp ml) was injected into the DMS, and fibers were placed on the cell bodies in VTA/SNc (M–L ± 1.4, A–P 0.74, D–V −2.6 mm) to enable recordings from retrogradely labeled cells (n = 4 mice). Three of the mice were recorded from both hemispheres, providing a total of n = 7 recording sites.

One mouse was used for the GFP recordings as a control condition for VTA/SNc::DMS terminals recordings (*Figure 1e*).

## Instrumental reversal learning task

The recordings were obtained while the mouse performed a reversal learning task in an operant chamber with a central nose poke, retractable levers on each side of the nose poke, and reward delivery in a receptacle beneath the central nose poke.

Each trial began with the illumination of the center nose port. After the mouse entered the nose port, the two levers were presented with a delay that varied between 0–1 second. The mouse then had 10 seconds to press a lever, otherwise the trial was classified as an abandoned trial and excluded from analysis (this amounted to <2% of trials for all mice). After the lever-press, an additional random 0–1 second delay (0.1 seconds intervals, uniform distribution) preceded either CS-with no reward delivery or CS+ with a 4 µl reward of 10% sucrose in $H_2O$. Reward outcomes were accompanied by different auditory stimul: 0.5 seconds of white noise for CS- and 0.5 seconds of 5 kHz pure tone for CS+. Every trial ended with a 3 seconds inter-trial delay (after the CS- auditory stimulus or the mice exit the reward port).

For the reversal learning, each of the levers either had a high probability for reward (70%) or low probability for reward (10%). Throughout the session, the identity of the high probability lever changed in a pseudorandom schedule; specifically, each block consisted of at least 10 rewarded trials plus a random number of trials drawn from a Geometric distribution of p=0.4 (mean 2.5). On average, there were 23.23 ± 7.93 trials per block and 9.67 ± 3.66 blocks per session. Both reported summary statistics are mean ± standard deviation.

## Data processing

All fiber photometry recordings were acquired at 15 Hz. 2–6 recording sessions were obtained per recording site (one session/day), and these recordings were concatenated across session for all analyses. On average, we had 1307.0 ± 676.01 trials per mouse (858.09 ± 368.56 trials per mouse for VTA/SN::DMS Terminals recordings and 448.91 ± 455.61 trials per mouse for VTA/SN::DMS Cell-bodies recordings).

The signals from each recording site were post-processed with a high-pass FIR filter with a pass-band of 0.375 Hz, stopband of 0.075 Hz, and a stopband attenuation of 10 dB to remove baseline fluorescence and correct drift in baseline. We derived dF/F by dividing the high-pass filtered signal by the mean of the signal before high-pass filtering. We then z-scored dF/F for each recording site, with the the mean and standard error calculated for the entire recording from each site.

The VTA/SN::DMS terminals data consisted of 10108 total trials across 12 recording sites, and VTA/SN::DMS cell-bodies consisted of 4938 total trials across 7 recording sites.

## Q learning mixed effect model

We fit a trial-by-trial Q-learning mixed effect model to the behavioral data from each of the 12 mice on all recording sites and combined data across mice with a hierarchical model. The model was initialized with a Q value of 0 for each action and updated at each trial according to:

$$Q_{t+1}(c_t) = Q_t(c_t) + \alpha(r_t - Q_t(c_t))$$

where $Q$ is the value for both options, $c_t$ is the option chosen on trial $t$ (lever either contralateral or ipsilateral to recording site), and $0 <= \alpha <= 1$ is a free learning rate parameter. The subject's probability to choose choice $c$ was then given by a softmax equation:

$$P(c_t = c) \propto exp(\beta \cdot Q_t(c) + stay \cdot I(c, c_{t-1}))$$

where $\beta$ is a free inverse temperature parameter, *stay* is a free parameter encoding how likely the animal will repeat its choice from the last trial, and $I$ is a binary indicator function for choice repetition (1 if $c$ was chosen on the previous trial; 0 otherwise). The three free parameters of the model were estimated separately for each subject, but jointly (in a hierarchical random effects model) with group-level mean and variance parameters reflecting the distribution, over the population, of each subject-level parameter.

The parameters were estimated using Hamiltonian Monte Carlo, as implemented in the Stan programming language (version 2.17.1.0; *Carpenter et al., 2017*). Samples from the posterior distribution over the parameters were extracted using the Python package PyStan (*Carpenter et al., 2017*). We ran the model with 4 chains of 1000 iterations for each (of which the first 250 were discarded for burn-in), and the parameter adapt_delta set to 0.99. We verified convergence by visual inspection and by verifying that the potential scale reduction statistic Rhat (*Gelman and Rubin, 1992*) was close to 1.0 (<0.003 for all parameters) (*Table 1*).

We used the sampled parameters to compute per-trial Q values for each action, trial, and mouse. We calculated the difference between the Q values for the chosen action and unchosen action for each trial. We binned the difference in these Q values for each trial and plotted the average GCaMP6f time-locked to lever presentation for each bin (*Figure 3b,e*).

## Regression model

In *Figure 3c,f*, we performed a linear mixed effect model regression to predict GCaMP6f signal at each time point based on Q-values, choice (contralateral vs ipsilateral), their interaction, and an intercept. We took the difference in Q values for the chosen vs unchosen levers, then we standardized the difference in Q values for each mouse and each recording site. GCaMP6f was time-locked to lever presentation, regressing to data points 1 second before and 2 seconds after the time-locked event for 45 total regressions. The regression, as well as the calculation of p values, was performed with the MixedModels package in Julia (*Bezanson et al., 2014*). The p values were corrected for false discovery rate over the ensemble of timepoints for each regression variable separately, using the procedure of Benjamini and Hochberg (*Benjamini and Hochberg, 1995*) via the MultipleTesting package in Julia (*Bezanson et al., 2014*).

## Multiple event kernel analysis

In *Figure 3—figure supplement 5*, we fit a linear regression model to determine the contributions to the ongoing GCaMP6f signal of three simultaneously modeled events (nose poke, lever presentation, lever press). To do this, we used kernels, or sets of regressors covering a series of time lags covering the period from 1 second before to 2 seconds after each event. Each event had four kernels, corresponding to the four conditions from *Figure 3a,c* (all combinations of contralateral vs ipsilateral trials and previous reward vs no previous reward trials). We solved for the kernels by regressing the design matrix against GCaMP6f data using least squares in R with the rms package (*Harrell, 2018*). The standard error (colored fringes) was calculated using rms' robcov (cluster robust-covariance) function to correct for violations of ordinary least squares assumptions due to animal-by-animal clustering in the residuals.

## Acknowledgments

We thank the entire Witten and Daw labs for comments, advice and support on this work. IBW is a New York Stem Cell Foundation—Robertson Investigator.

## Additional information

### Funding

| Funder | Grant reference number | Author |
| --- | --- | --- |
| National Institute for Health Research | 5R01MH106689-02 | Ilana B Witten |
| New York Stem Cell Foundation | Robertson Investigator | Ilana B Witten |
| Army Research Office | W911NF-16-1-0474 | Nathaniel D Daw |
| Army Research Office | W911NF-17-1-0554 | Ilana B Witten |

The funders had no role in study design, data collection and interpretation, or the decision to submit the work for publication.

### Author contributions

Rachel S Lee, Conceptualization, Software, Formal analysis, Validation, Investigation, Methodology, Writing—original draft, Writing—review and editing; Marcelo G Mattar, Investigation, Methodology; Nathan F Parker, Data curation, Investigation, Methodology; Ilana B Witten, Conceptualization, Resources, Data curation, Supervision, Funding acquisition, Validation, Methodology, Writing—original draft, Writing—review and editing; Nathaniel D Daw, Conceptualization, Formal analysis, Funding acquisition, Investigation, Methodology, Writing—original draft, Writing—review and editing

### Author ORCIDs

Rachel S Lee (ID) http://orcid.org/0000-0001-7984-1942
Ilana B Witten (ID) http://orcid.org/0000-0003-0548-2160
Nathaniel D Daw (ID) https://orcid.org/0000-0001-5029-1430

### Decision letter and Author response

Decision letter https://doi.org/10.7554/eLife.42992.025
Author response https://doi.org/10.7554/eLife.42992.026

## Additional files

### Supplementary files

• Source code 1. Mixed Effect Q-learning Model Code. Stan code for the trial-by-trial Mixed Effect Q-learning Model (more details in Materials and methods section).
DOI: https://doi.org/10.7554/eLife.42992.016

• Source code 2. Regression Model for *Figure 3c,f*. Julia code for running the regression model of GCaMP6f against Q values, mice's action, the interaction between those two variables, and an intercept. Regression was performed using Julia's MixedModels package. See 'Regression Model' in Materials and methods section for more details.
DOI: https://doi.org/10.7554/eLife.42992.017

• Source data 1. GCaMP6f data. Each row is one timepoint, with columns that denote the GCaMP signal for that timepoint, binary indicator variables for behavioral events at that timepoint (0 represents no event at this timepoint, 1 represents event occurred at this timepoint), the recording site, session, and high value lever at the timepoint. Behavioral events include trial start, nose poke enter and exit, lever presentations, and contralateral or ipsilateral lever press.
DOI: https://doi.org/10.7554/eLife.42992.018

• Source data 2. Trial-by-trial data with Q values and GCaMP6f. CSV with relevant trial information for each trial across terminals and cell-bodies data. Trial information includes the recording location, recording site ID, session ID, the mouse's choice, and whether or not the mouse was rewarded. Additional columns include the Q values for each trial (including Q value of contralateral minus ipsilateral choice and Q values of chosen minus unchosen choice) and the z-scored GCaMP signal time-

locked at four important behavioral events (nose poke, lever presentation, lever press/choice, and reward).

DOI: https://doi.org/10.7554/eLife.42992.019

• Transparent reporting form

DOI: https://doi.org/10.7554/eLife.42992.020

## Data availability

All data generated or analysed during this study are included in the manuscript and supporting files.

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
