## [Decision Letter]

Thank you for submitting your article "Value representations do not explain movement selectivity in DMS-projecting dopamine neurons" for consideration by *eLife*. Your article has been reviewed by three peer reviewers, including Geoffrey Schoenbaum as the Reviewing Editor and Reviewer #1, and the evaluation has been overseen by Timothy Behrens as the Senior Editor. The following individual involved in review of your submission has agreed to reveal their identity: Ingo Willuhn (Reviewer #3).

The reviewers have discussed the reviews with one another and the Reviewing Editor has drafted this decision to help you prepare a revised submission.

Summary:

This study involves a reanalysis of data published previously (Parker et al., 2016), examining signaling in DMS-projecting dopamine neurons in mice performing a probabilistic reversal task. Bulk Ca++ signaling was recorded from cell bodies and terminals in DMS using fiber photometry, and activity was examined for correlates of RPE's versus movement direction. In the new analyses, the authors show in more detail than previously that these dopamine neurons carry a movement or action related signal in addition to the RPE signal.

Essential revisions:

The reviewers agreed that the new results provide potentially important new information regarding movement-related dopaminergic correlates. Generally however each reviewer had some difficulties following methodological details necessary to fully evaluate the new data. While the details differ across the reviews, the general issue was a lack of clarity regarding what was analyzed. The additional details might also include additional analyses looking at data before the lever press or versus baseline (reviewer 2) and analyses showing that the relatively high remaining probability for contralateral lever presses at 0.35-0.4 (Figure 1B) is not a problem (first point, reviewer 3). Finally, the reviewers also felt that more needed to be done in the Introduction and Discussion to make clear what was new here, and also in the Discussion to relate the current findings to other data that has been presented regarding movement correlates.

*Reviewer #1:*

In the current study the authors examine dopamine signaling in DMS projecting midbrain neurons in mice performing a probabilistic reversal task. Bulk Ca++ signaling was recorded from cell bodies and terminals in DMS using fiber photometry, and activity was examined for correlates of RPE's versus movement direction. The authors report that both signals were observed in both locations. They conclude that dopamine neurons carry a movement or action related signal in addition to the RPE signal. The results are of potential significance given the historical involvement of dopamine in movement function, which has been eclipsed by the error signaling function in recent years, along with the increasing evidence that the RPE hypothesis does not fully explain phasic dopaminergic signaling. Finding of action correlates, particularly if they could be argued to be action-related error signals, would be quite interesting. That said I have a conceptual question and several methodological concerns.

Conceptually, I think what is most interesting is the possibility this is an error-related signal that is simply not in the value domain. This possibility is alluded to in the Discussion. But not much is said there, nothing is said earlier, and it is unclear to me what evidence there is for this? Did I understand this right? Is there evidence? If I did and there is not, can the authors expand on this speculation and what experiment would show it? This is very important because otherwise it is not clear to me where this study goes beyond other studies, such as the prior Witten report or the one by Walton. A paragraph reviewing and contrasting this result with those would help also.

Methodologically, I am concerned because I may not be following where the trials are coming from with the sparse design. The mice are performing a choice task in which there is a high value on one side and low value on the other. The location of the high value option switches frequently, it looks like after 40ish trials. And on each trial the mouse is free to go in either direction. As a result, most of the responses in one direction are early in a block, whereas most of the responses in the other direction are late in the block. There are no forced trials, so this results in a dramatic asymmetry in where the relevant data comes from in a block it seems to me. In other words, in any block the comparison is largely between trials in one direction early in learning (or before) and the other mostly late (after) learning. Further, the comparison is made between trials after a rewarded trial and trials after a non-rewarded trial. Given the different probabilities, if the directions are not segregated, then there will be an asymmetry here also, since there will be many more trials after reward on the 70% reward schedule and many more trials after non-reward on the 10% reward schedule. Of course, I realize the authors know this, but the manuscript does not explain well how this is handled. If possible, I'd like a clean comparison of trials matched for the stage of task to show the effect. If this is not possible, then if the shape of the data can be made more clear and how these issues are handled, that might be sufficient. But a naïve reader who is not deeply familiar with the task, as the authors are, needs to be able to understand where the trials are coming from. At present, I could not do this.

*Reviewer #2:*

This study involves a deeper analysis of a previously published data set from the Witten group concerning the correlates of dopamine axonal activity in dorsomedial striatum. The previous paper reported direction specific effects in DMS (but not ventral striatal) axons and DMS-projecting cell bodies; the current study investigates whether the DMS dopamine represents a signal more closely aligned to the relative value of making an ipsilateral v contralateral action or the value of the chosen response (interacting with action). The authors' analyses point towards a conclusion that the dopamine signal is shaped by the value of the chosen action and by the direction of movement.

The original finding of a direction specific dopamine response was already interesting, and this study certainly finesses that result in a clean manner. However, I was not entirely convinced of how much this really advances from the original finding to tell us what dopamine is doing.

The different predictions are nicely set out in Figure 2 (though it might be best not to include the "chosen value modulation" option here given it is already a non-starter for DMS dopamine based on the Parker data set) and one model is clearly supported based on the data are presented in Figure 3. But what the authors focus on – is the dopamine best described as RPE x action or chosen value x action – struck me as rather small scale, particularly given there is much more evidence for dopamine encoding chosen value in some form. While I found this an interesting conclusion, it seemed hardly like it would really help advance the ongoing and passionate RPE v movement debates.

Moreover, it appears as if there is a lot more in these data than is remarked upon. For instance, there appears already to be a meaningful relationship between dopamine activity and the animal's upcoming action in the pre-lever period. Indeed, at least in the cell bodies, if you account for this baseline shift – and what the baseline is in these analyses was never clearly defined for me – the phasic action component looks like it would be much weaker at the time of lever extension. This interaction between timescales is worth considering and commenting on in more detail, particularly in the light of the Hamid/Berke findings that what can look like an RPE when baselined pre-event of interest might look very different if a baseline is taken at an earlier timepoint. Another important idea that was not specifically addressed was whether dopamine activity reflects the reward prediction or the vigour of the (contralateral) action. Is there enough variance between the Q value and, say, initiation speed to include that as an additional regressor? The chosen minus unchosen value signals come pretty late given the speed of the GCaMP6f, so what is actually driving these here?

*Reviewer #3:*

The authors investigated the activity of dopamine neurons that project from the midbrain to the dorsomedial striatum (DMS) during a probabilistic instrumental reversal-learning task in mice using fiber photometry for calcium imaging of both neuron terminals in the DMS and cell bodies in the midbrain. Specifically, they explored dopamine neuron activity during the time when choices were made towards an operandum of the Skinner box located contralaterally to the recording site in the brain. This data had been published in Nature Neuroscience previously (Parker et al., 2016), but in that publication the authors had not studied the reported modulation of DMS-projecting neurons by contralateral choices in as much depth as they do here. Here, the authors aimed at determining whether these signals are related to movement or contralateral reward prediction errors (RPEs). The authors report that dopamine neuron activity modulated by contralateral choices is distinct from RPEs, which according to them implies that it is better explained by movement direction.

The topic of this study is of great interest to the fields of behavioral and computational neuroscience, as the mechanisms by which regional differences in dopamine signaling contribute to behavioral flexibility are still not understood. The authors conducted sophisticated and computationally challenging analyses that deliver highly interesting findings. It speaks for their thoroughness that results were assessed on both the level of terminals and cell bodies. Furthermore, experimental design and data presentation are sound, and the manuscript is well-written. However, I have a few concerns:

- A remaining probability for contralateral lever presses at 0.35-0.4 (Figure 1B), 7-10 trials after the ipsilateral lever has become the high-probability option, seems quite high. Especially, since probability for choosing the contralateral lever, when it is the high-probability option, gets to around 0.9. Is the animals' behavior towards the two sides comparable? Is there a bias? This is essential for the analyses performed (e.g., if the number of rewarded trials is different, interpreting how trial history affects activity becomes more difficult). The authors need to both test and discuss this.

- Can the authors exclude that the position of the optic fiber on the skull (and attached equipment; above left or right hemisphere) contributed to contralateral movements being different in their execution compared to ipsilateral movements? In other words, did implanting on one side of the skull influence the animals' balance or their ability to move in any direction due to tethering or did animals' heads tilt towards or away from the implant (due to weight or torque)? A photo of the setup including a connected animal performing in the task may prove useful in this context.

- The authors frequently refer to movement signals. Can the authors distinguish between movement and motivation?

- Does the contralateral movement-related calcium signal correlate with lever-press latency (on a trial-by-trial basis)?

- In the Discussion, the authors should speculate on how unilateral dopamine neuron signals affect the contralateral side of the body (e.g., limbs or else) in order to initiate/support/perform a movement. This is a central part of the conclusion, if I am not mistaken, and should be honored with a speculation on how this may be implemented in terms of functional neuroanatomy. Also, rotation behavior after 6-OHDA lesion should be addressed in this context.

- In the Materials and methods section, it is stated that 1-5 recordings were obtained per recording site. Does that mean that some animals contributed a lot more data than others? For example, 10.108 "terminal" trials were recorded. That makes about 840 per animal on average. Is that roughly the average number of trials per animal? If not, it should be reported.

---

## [Author Response]

Essential revisions:The reviewers agreed that the new results provide potentially important new information regarding movement-related dopaminergic correlates. Generally however each reviewer had some difficulties following methodological details necessary to fully evaluate the new data. While the details differ across the reviews, the general issue was a lack of clarity regarding what was analyzed. The additional details might also include additional analyses looking at data before the lever press or versus baseline (reviewer 2) and analyses showing that the relatively high remaining probability for contralateral lever presses at 0.35-0.4 (Figure 1B) is not a problem (first point, reviewer 3). Finally, the reviewers also felt that more needed to be done in the Introduction and Discussion to make clear what was new here, and also in the Discussion to relate the current findings to other data that has been presented regarding movement correlates.

Thank you so much for the feedback. We have included more details on the set up and structure of the reversal learning task in order to help clear up any confusion on the methodological details.

We addressed reviewer 2’s request with additional analyses on the data time-locked to the nose poke event and also using a multiple event regression that does not a priori choose which event to time-lock the GCaMP6f. Our regression model instead characterized the response as arising from contributions linked to each task event (nose poke, lever presentation, lever press), each captured with a separate kernel. We showed that this analysis supports our original results.

In addition, we performed additional analyses showing that the mice did not have a direction or reward bias before or after a block switch. For reviewer 3’s specific concern, we showed that the higher preference for the contralateral side is not an issue: the way the data was depicted previously gave a misleading impression, but overall the behavior around the time of both types of block switches (contralateral to ipsilateral and vice versa) is very similar.

Finally, we expanded our literature review in our Introduction and Discussion to include relevant findings, how they relate to our results, and how our results are novel in comparison. We appreciate the reviewers’ helpful comments, which helped us greatly improve our manuscript with further analyses that helped solidify our results.

We included a number of additional figures and results in this letter; although we are open to advice on this point, we have not included some of them in the supplementary material for the paper because, should the paper be accepted, we understand the response would also be published.

Reviewer #1:[…] Conceptually, I think what is most interesting is the possibility this is an error-related signal that is simply not in the value domain. This possibility is alluded to in the Discussion. But not much is said there, nothing is said earlier, and it is unclear to me what evidence there is for this? Did I understand this right? Is there evidence? If I did and there is not, can the authors expand on this speculation and what experiment would show it? This is very important because otherwise it is not clear to me where this study goes beyond other studies, such as the prior Witten report or the one by Walton. A paragraph reviewing and contrasting this result with those would help also.

Thank you for this comment, and in particular for bringing up the Walton study, which leaves open very similar interpretational questions as the ones we address from Parker’s study. The Parker and Walton studies each report correlates in DA activity of both value expectation and of action identity. Whereas Parker shows that DA activity in DMS is elevated for contralateral choices; Walton shows that NAcc DA activity is elevated for “go” responses relative to “no-go”. While both of these articles show in different ways that the activity *also* reflects more conventional RPEs related to *overall* reward expectancy (i.e. PEs in state values V(s), which are sensitive in Walton’s case to things like cues about reward size), both leave open a central interpretational question about the action-related responses that is, to our knowledge, first clearly identified and also first decisively addressed in the current study.

This interpretational question is whether the apparently action-related responses are truly related to the action identity in a categorical sense, or whether they might actually instead be, in effect, artifacts of action-specific error signals (e.g., PEs for action values Q(s,a), sensitive to predictions about the value of a particular action in a situation). In Parker’s case, if DMS selectively processes value for contralateral movements, then elevated activity may be seen, on average, on trials when the contralateral action is chosen, because those also tend to be trials when that action predicts relatively higher value and RPEs for contralateral movements are positive. In Walton’s case, NAcc might analogously represent the value of “go” (relative to “no-go”) responses, producing greater responding when “go” (relative to “no-go”) responses are valued: again, on average, those trials and conditions when “go” is correctly chosen. This hypothesis is plausible given NAcc DA’s role in appetitive approach. (Parkinson et al., 2002). In any case, we were able to decisively rule out this concern in Parker’s data by articulating the distinction between chosen value and action value and showing that DMS DA follows the former. Although given hindsight and the conceptual advances in the current paper, we can observe hints of a similar distinction in the Walton results (and in the current revision we discuss how these help to buttress our interpretation), we would stress that almost all the significant action-related value results in the Walton paper (with the exception of a single time point in Supplementary Figure 9E) concern the modulation of “go”-related activity by reward expectancy (consistent with both chosen and action value), and so do not directly test or address what we identify as the key differentiating question of whether this reverses for “no-go” responses.

Why is all this important, and what does it have to do with the reviewer’s question about error signals for motor responses? The reports of activity apparently related to movement per se are important both in a positive sense (because they suggest a function more directly related to movement elicitation or control, as Walton and Parker both point out and we also now say more clearly); yet at the same time they are deeply puzzling in that it is difficult to understand how they can be reconciled with the substantial evidence for RPE signaling–how, for instance, can recipient structures distinguish the error-related components of the signal that should control plasticity vs. the interleaved movement-related ones that should initiate actions?

Against this background, one of the main conceptual advances of our article was to articulate clearly, and then show how to test definitively, a way in which the RPE and motor responses could have been reconciled: specifically, we posited that DMS carries an RPE for action value that would account for both responses. In fact, having set up and tested this possibility, we end up rejecting it: This strengthens the (still important and still puzzling) case for truly movement-related signaling. That said, we agree with the reviewer that our closing suggestion about whether this signal is a truly movement- rather than value-related error is indeed a novel conceptual advance, although not the main one of the paper. Our point (which we have tried to clarify and elaborate) is that another possibility is that the movement direction signal whose existence we verify might be useful as different a sort of error signal, for training a class of S-R habit models that goes back to Guthrie, E.R. (Guthrie, 1935), and has recently been rediscovered. But this is more in the category of interpretations left open by the current study. We do not as yet have direct evidence bearing on this point either way, and it remains for future work (probably using causal manipulations) to address it.

Methodologically, I am concerned because I may not be following where the trials are coming from with the sparse design. The mice are performing a choice task in which there is a high value on one side and low value on the other. The location of the high value option switches frequently, it looks like after 40ish trials. And on each trial the mouse is free to go in either direction. As a result, most of the responses in one direction are early in a block, whereas most of the responses in the other direction are late in the block. There are no forced trials, so this results in a dramatic asymmetry in where the relevant data comes from in a block it seems to me. In other words, in any block the comparison is largely between trials in one direction early in learning (or before) and the other mostly late (after) learning. Further, the comparison is made between trials after a rewarded trial and trials after a non-rewarded trial. Given the different probabilities, if the directions are not segregated, then there will be an asymmetry here also, since there will be many more trials after reward on the 70% reward schedule and many more trials after non-reward on the 10% reward schedule. Of course, I realize the authors know this, but the manuscript does not explain well how this is handled. If possible, I'd like a clean comparison of trials matched for the stage of task to show the effect. If this is not possible, then if the shape of the data can be made more clear and how these issues are handled, that might be sufficient. But a naïve reader who is not deeply familiar with the task, as the authors are, needs to be able to understand where the trials are coming from. At present, I could not do this.

Thank you for this comment. As we understand it, the reviewer raises a family of potential concerns that the data underlying either or both factors in the 2x2 (rewarded/non by ipsi/contra) in Figure 3A, D might be incomparable due to tending to arise at different times in the progression of learning and relearning given the free-choice, frequently reversing design. We have examined this concern in a number of ways.

First, in general, we would note that any imbalance isn’t especially severe. First, the animals adapt to value changes fairly rapidly, favoring the new best lever within a few trials, as shown in the updated Figure 1B. Given that the average block length is 23.23 ± 7.93 trials per block (minimum, 12; n = 19 recording sites across both terminals and cell-bodies data), the majority of the data are collected during periods when the choices favor whichever lever is currently best. This figure also indicates that even asymptotically, choices aren’t especially exclusive to the high-value side; there is decent sampling of both options.

We should also point out that there are many reversals per session (mean +/ sd: 8.67 ± 3.66, as is now reported in the revised manuscript); thus ipsilateral and contralateral each serve as both high and low value options many times, and are fully counterbalanced in this respect. Thus, in particular, any imbalances with respect to early and late sampling of (or to rewarded and nonrewarded sampling from) high vs. low value levers are counterbalanced with respect to their relationship with the factor of interest, ipsi vs. contra. Finally, although the reviewer doesn’t raise it explicitly here, there might be an additional, related concern that some animals tend to favor one lever (e.g., the contralateral one) overall, leading to another possible avenue for unbalanced sampling. However, as we discuss below at several points in the response to reviewer 3, this also turns out not to have a noticeable effect, as these biases are small and not consistent, and we have data from a subset of animals obtained from both sides simultaneously.

Finally, to ensure more directly that our results are not affected by whether choices occurred early versus late in a block, we repeated the analysis by splitting the trials to early (first 6 trials in the block) and late trials (last 6 trials in the block) (Author response image 1; Author response image 2 The shortest length for a block was 12 trials, so we used only the first 6 and last 6 trials from each block to ensure all blocks contributed equally to the averaged traces. (For the plots depicting activity by ranges of Q values, we also split them into four bins due to the smaller amount of data.)

Note that we see similar results in both VTA/SN::DMS terminals and VTA/SN::DMS cell-bodies even after splitting the data into early vs. late in the blocks as we did for Figure 3. In particular, we see that the signals are modulated by contralateral movement and whether or not the mice were rewarded previously. We also see similar results when breaking out responses by Q values. As in the main analysis, the regression results indicate that there are some significant effects for contralateral action and difference in Q values for chosen and unchosen, but no significant effect for the interaction between the two. Thank you again to the reviewer for suggesting this additional analysis to confirm that our results still hold even taking into account the stage of relearning.

**Author response image 1. respfig1:** Early and late trials in block are both modulated by chosen value and contralateral action (VTA/SN::DMS Terminals, n = 12 sites) (**A**) GCaMP6f signal from VTA/SN::DMS Terminal (n = 12 sites) from first 6 trials of each block. Traces are time-locked to the lever presentation for contralateral trials (blue) and ipsilateral trials (orange), as well as rewarded (solid) and non-rewarded previous trial (dotted). Colored fringes represent 1 standard error from activity averaged across recording sites (n = 12) (**B**) GCaMP6f signal for contralateral trials (blue) and ipsilateral trials (orange), and further binned by the difference in Q values for chosen and unchosen action. Colored fringes represent 1 standard error from activity averaged across recording sites (n = 12). (**C**) Mixed effect model regression on each datapoint from 3 seconds of GCaMP6f traces. Explanatory variables include the action of the mice (blue), the difference in Q values for chosen vs. unchosen actions (orange), their interaction (green), and an intercept. Colored fringes represent 1 standard error from estimates. Dots at bottom mark timepoints when the corresponding effect is significantly different from zero at p<.05 (small dot), p<.01 (medium dot), p<.001 (large dot). P values were corrected with Benjamini Hochberg procedure. (D-F) Same as (A-E), except using the last 6 trials of each block.

**Author response image 2. respfig2:** Early and late trials in block are both modulated by chosen value and contralateral action (VTA/SN::DMS Cell-bodies, n = 7 sites) (**A**) GCaMP6f signal from VTA/SN::DMS Cell-bodies (n = 7 sites) from first 6 trials of a block. Traces are time-locked to the lever presentation for contralateral trials (blue) and ipsilateral trials (orange), as well as rewarded (solid) and non-rewarded previous trial (dotted). Colored fringes represent 1 standard error from activity averaged across recording sites (n = 7). (**B**) GCaMP6f signal for contralateral trials (blue) and ipsilateral trials (orange), and further binned by the difference in Q values for chosen and unchosen action. Colored fringes represent 1 standard error from activity averaged across recording sites (n = 12). (**C**) Mixed effect model regression on each datapoint from 3 seconds of GCaMP6f traces. Explanatory variables include the action of the mice (blue), the difference in Q values for chosen vs. unchosen actions (orange), their interaction (green), and an intercept. Colored fringes represent 1 standard error from estimates. Dots at bottom mark timepoints when the corresponding effect is significantly different from zero at p<.05 (small dot), p<.01 (medium dot), p<.001 (large dot). P values were corrected with Benjamini Hochberg procedure. (D-F) Same as (A-E),except using trials from the last 6 trials of the block.

Reviewer #2:[…] The original finding of a direction specific dopamine response was already interesting, and this study certainly finesses that result in a clean manner. However, I was not entirely convinced of how much this really advances from the original finding to tell us what dopamine is doing.The different predictions are nicely set out in Figure 2 (though it might be best not to include the "chosen value modulation" option here given it is already a non-starter for DMS dopamine based on the Parker data set) and one model is clearly supported based on the data are presented in Figure 3. But what the authors focus on – is the dopamine best described as RPE x action or chosen value x action – struck me as rather small scale, particularly given there is much more evidence for dopamine encoding chosen value in some form. While I found this an interesting conclusion, it seemed hardly like it would really help advance the ongoing and passionate RPE v movement debates.

Thank you for this comment. We chose to include the “chosen value modulation” option for pedagogical reasons, because it was helpful to first introduce the idea of “chosen value modulation,” then include the contralateral action modulation on top of that. The second theory helps readers understand the two types of modulations involved in the third theory.

We also appreciate the opportunity to do a clearer job explaining the contributions of our study. We have attempted to sharpen these points in the current revision. (We would also direct your attention to the first reviewer’s first comment for more discussion about this.) The discussion on how to reconcile dopamine’s involvement in reward vs. movement is perhaps the single central puzzle in the study of this neuromodulatory system going back decades to the initial discoveries of its involvement in self-stimulation and disorders of movement. One of the reasons for the excitement surrounding the RPE theories was that they seemed to offer a detailed, quantitative (though of course stylized) way to reconcile these views. However, recent reports (among them ours) of seemingly motor-related responses that are apparently distinct from RPEs have recently reopened the classic questions, and–given the substantial evidence for the RPE account– introduced new questions, such as how recipient structures could possibly distinguish interleaved RPE and movement signals with different functions.

A chief contribution of the current study is to articulate a proposal for how the movement-related responses might be interpreted in the RPE framework, by extending it to include action RPEs. The action value vs. chosen value distinction is central for posing, testing, and ultimately rejecting this possibility. While it is true that we end up restoring Parker’s conclusion, we now know more about this important issue–both in the substantive sense that we have identified and closed interpretational ambiguities in Parker’s (and other) results, and in laying conceptual groundwork that will be relevant going forward.

In particular, as we now say in the revised article, we believe that our basic framework and approach will be relevant in confronting other aspects of the growing body of evidence that DA signals may encode for variables beyond RPE. Recent studies, for instance, showed that midbrain DA neurons may also encode behavioral variables relevant to the task, such as the mice’s position, their velocity, their view-angle, and the accuracy of their performance (Howe et al., 2013; da Silva et al., 2018; Engelhard et al., 2018). Our modeling provides a framework for understanding how these DA responses can be interpreted in different reference frames and how they might ultimately encode some form of RPEs with respect to different behavioral variables in the task. Even though this turned out not to be the case for Parker’s results, it may well apply elsewhere. This conceptual framework can be extended to help understand the heterogeneous DA responses from more complicated real-world, high-dimensional reinforcement learning tasks.

Moreover, it appears as if there is a lot more in these data than is remarked upon. For instance, there appears already to be a meaningful relationship between dopamine activity and the animal's upcoming action in the pre-lever period. Indeed, at least in the cell bodies, if you account for this baseline shift – and what the baseline is in these analyses was never clearly defined for me – the phasic action component looks like it would be much weaker at the time of lever extension. This interaction between timescales is worth considering and commenting on in more detail, particularly in the light of the Hamid/Berke findings that what can look like an RPE when baselined pre-event of interest might look very different if a baseline is taken at an earlier timepoint.

Thank you for this comment. We did not baseline our responses when analyzing the GCaMP6f time-locked to the lever presentation, and we agree it is important to better understand the extent to which the key effects are already present in the signal at earlier timepoints and to what extent taking this into account changes the picture at the time of lever presentation. We now include a number of additional analyses examining these issues, which in general do not change our overall conclusions.

First, we found the same basic pattern of effects when we aligned signals to the nose poke event, in an analysis which we have now included as Figure 3—figure supplement 4. Just as in Figure 3, we see clear modulation by chosen value and contralateral action when we break down signals both by previous reward and action (Figure 3—figure supplement 4A) and by Q values and action (Figure 3—figure supplement 4B). The regression results in Figure 3—figure supplement 4C indicate that the signals were significantly modulated by the contralateral action and Q values. Although (especially in the cell bodies results in Figure 3—figure supplement 4D-F) there appears to be a distinct component of response time-locked to the nose poke, the bulk of the response is more smeared out and at higher latency, such that the significant effects of both action and value actually occur shortly following the mean time of lever presentation (denoted by the black diamond with a line indicating the range containing 80% of latency values). All this suggests the modulation of DA signals is more closely related to lever presentation. As before, we see similar effects for both VTA/SNc::DMS terminals and VTA/SNc::DMS cell bodies (Figure 3—figure supplement 4D-F).

Finally, to more directly verify that our conclusions are independent of baseline effects and of responses to the other events, we also modeled the GCaMP6f signals independent of the time-locked event. This approach, in effect, takes account of any baseline signal related to other events, which we believe is more flexible, and more interpretable, than subtracting off a single baseline arbitrarily defined at some other time point. In particular, we performed a multiple regression with response kernels capturing the contribution of components linked to each of the three time-locked events simultaneously. To parallel the analysis from Figure 3A, D, we included, for each event, a kernel (i.e. a series of time-lagged regressors, covering timesteps from 1 second before until 2 seconds after each event) for each combination of action (contra or ipsi), and previous reward (or none). We estimated all the effects simultaneously using least-squares regression, thereby trading off the responsibility of the different events in explaining components of the signal (see Materials and methods subsection “Multiple event Kernel Analysis” for more details). The resulting output is kernels for each of the time-locked events and for each of the four conditions. We included these results as part of Figure 3—figure supplement 5

Although weaker (due to dividing variance up among many more explanatory variables), these analyses basically recapitulate the results of the simpler peri-event analyses. The key ipsi-contra separation is visible in all three kernels, and the direction of the reward effect is consistent as well, with at least a trend toward higher signal following non-reward than reward consistently across both ipsi and contra trials and across most of the kernels. Although this analysis does not entirely attribute the effect to any single event (and this may either reflect that the data really do arise from multiple effects time-locked to different events, or a failure of least squares regression and the linear convolutional model to completely identify an actually isolated effect), the sharp phasic signal to the lever presentation remains similar to the initial time-locked analysis, despite the portion of the effect taken by the other events. Note also that the lever press kernels in Figure 3—figure supplement 5C verify the same clear crossing effect that occurs right after the mice press the lever, as also noted in Figure 4.

Finally, we also performed a multiple event regression examining the effects of action value as a continuous variable, to parallel to the results from Figure 3C, F. In this case, we include time-lagged regressors (kernels) for intercept, contralateral action, Q values for each trial, and the interaction between Q values and contralateral action for each kernel. Again, we solved for the regressors simultaneously using least-squares regression. As before, we calculated p values (corrected with Benjamini Hochberg procedure) to determine when the regressors’ effects were significantly different from zero.

**Author response image 3. respfig3:** Kernels for each significant behavioral event for mixed effect model regression (**A**) Nose poke kernel output from linear regression model using GCaMP6f from VTA/SN::DMS terminals. Each line represents a normalized regression variable: action (blue; 0 for ipsilateral, 1 for contralateral), difference in Q values for chosen direction and unchosen direction (orange), and the interaction between the two (green). Colored fringes represent 1 standard error from activity averaged across recording sites (n = 12). Black diamond represents the average latency for lever presentation from nose poke with the error bars showing the spread of 80% of the latency values. (**B**) Lever presentation kernels, with the black diamond representing the average latency from lever press to lever presentation. (**C**) Lever press kernels, with the black diamond representing the average latency from CS+ or CS- to lever press. (D-F) Same as (A-E), except with signals from VTA/SN::DMS cell bodies averaged across recording sites (n = 7) instead of terminals.

As before, we see significant effects of contralateral action in all three kernels (Author response image 3). Significant positive modulation by chosen Q value is still seen primarily in the lever presentation and lever press kernels (Author response image 3). As with in Figure 3C, F, we do not see a significant effect from the interaction terms, suggesting that value effects reflect chosen value rather than side-specific value. In the lever press kernels (Author response image 3), we again see the contralateral action regressors cross from positive to negative soon after the lever press, reaffirming results from Figure 4. We see similar results in VTA/SN::DMS cell-bodies recordings, though the effect is weaker in the lever presentation kernels (Author response image 3).

Another important idea that was not specifically addressed was whether dopamine activity reflects the reward prediction or the vigour of the (contralateral) action. Is there enough variance between the Q value and, say, initiation speed to include that as an additional regressor? The chosen minus unchosen value signals come pretty late given the speed of the GCaMP6f, so what is actually driving these here?

Thank you for this important question. We considered the lever-press latency as a measure of vigor as we did not have video or other measures of vigor available. To investigate this issue we redid the regression analysis in Figure 3C, F but included the latency of lever press as an additional nuisance covariate (Figure 3—figure supplement 7). Our results indicated that latency of the lever press was not a strong predictor for GCaMP6f signals. Our conclusions with regards to the original variables remained the same. In order also to address the reviewer’s parenthetical note that the DA activity might reflect the vigor of *contralateral* action specifically, we also repeated this analysis on only contralateral trials (not shown). As with the results, the latency of lever press was still not a strong or significant predictor for GCaMP6f signals on contralateral choice trials. We’d like to thank the reviewer for suggesting this additional analysis, which confirmed that the DA activity was related to both chosen value and contralateral choice, unconfounded by response vigor insofar as we can estimate it.

Reviewer #3:[…] - A remaining probability for contralateral lever presses at 0.35-0.4 (Figure 1B), 7-10 trials after the ipsilateral lever has become the high-probability option, seems quite high. Especially, since probability for choosing the contralateral lever, when it is the high-probability option, gets to around 0.9. Is the animals' behavior towards the two sides comparable? Is there a bias? This is essential for the analyses performed (e.g., if the number of rewarded trials is different, interpreting how trial history affects activity becomes more difficult). The authors need to both test and discuss this.

Thank you for this important question. We apologize that the plot originally shown in the paper gave a misleading impression, which we discuss below. But first, on examining the preferences of the mice overall, we did not find that they strongly or consistently preferred the contralateral or ipsilateral action.

On average, the mice preferred the contralateral action 53.07% ± 9.73 (averaged across n =19 recording sites for terminals and cell-bodies data): any side bias was weak and not consistent from animal to animal. In response to another one of your questions, we also presented data from both hemispheres in a subset of animals we recorded from the same recording site bilaterally (Figure 3—figure supplement 6). The activity from each hemisphere still favored the contralateral choice, showing that the effect is not some accident of animals favoring one side or the other.

Regarding Figure 1B, we apologize for arbitrarily depicting the reversals as a switch from contralateral to ipsilateral as the high-value lever. This gave a misleading impression that there was a ipsi vs. contra bias, when the difference in responding simply reflected the progression of relearning (from late in the block on the left of the plot, to early in the block on the right). In fact, when we repeat the analysis to depict block switches from both contralateral to ipsilateral and vice versa, the results are very similar:

Note that both plots show that, by the time of a switch, reasonably high preference had developed for whichever lever was serving as high value, followed by gradual relearning after the reversal. Our own impression is this adjustment is pretty nimble, and a bit of probability matching rather than really exclusive focus on the better lever is not too surprising. In any case, these plots clearly reflect the difference in behavior between before and after the switch, not between ipsi and contra. This further shows that the higher probability for contralateral lever is part of the mice’s behavior during switch transitions, and not an indication of some choice bias. Since there was no difference between sides (and we had not intended to suggest one) we updated Figure 1B to average over both types of switches in a single plot. Thank you again to the reviewer for noticing the potential asymmetry that pointed us to the additional analysis that clarified the mice’s behavior.

- Can the authors exclude that the position of the optic fiber on the skull (and attached equipment; above left or right hemisphere) contributed to contralateral movements being different in their execution compared to ipsilateral movements? In other words, did implanting on one side of the skull influence the animals' balance or their ability to move in any direction due to tethering or did animals' heads tilt towards or away from the implant (due to weight or torque)? A photo of the setup including a connected animal performing in the task may prove useful in this context.

Thank you for this question. We did not implant on one side– all mice were implanted bilaterally to help with symmetry and balance. (For the DMS terminal animals, the second site was in the nucleus accumbens, and not analyzed in the current study.) The implant did not lead to any visible unbalance that we think could favor one direction of movement over the other. Consistent with that, in our nucleus accumbens recordings in our previous paper, we did not observe an overall contralateral bias in neural activity in DA terminals (Parker et al., 2016).

- The authors frequently refer to movement signals. Can the authors distinguish between movement and motivation?

Thank you for this question, which is subtle and thought-provoking. To clarify, when we described “movement signals,” we meant signals specific to the movement direction. Of course, as the reviewer points out below, there is reason to speculate this activity is well positioned to participate in the execution of contralateral movements; however, from mainly correlational data we cannot speak definitively about the function of these signals, and, in particular, we cannot and do not intend to rule out that these side-specific signals are related to functions like planning or monitoring of a lateralized movement, rather than movement execution per se. As for motivation, of course this is a broad term, but to some extent the central premise of our study is interrogating one version of this distinction. In particular, we distinguish whether the signals are best explained as related to the lateralized choice direction per se, or instead by the value that we estimate the animals attribute to that action. The underlying question here is precisely whether the seemingly side-specific responses are in fact instead related to the degree to which the animals are drawn to the action. We are mindful that the term “motivation” might have many meanings and some are difficult to pin down, but we do think the action value is one useful way to operationalize an aspect of it. Thus we conclude the activity is not related to motivation in this sense. We have added a brief comment on these issues to the Discussion.

- Does the contralateral movement-related calcium signal correlate with lever-press latency (on a trial-by-trial basis)?

Thank you for this important question. We address this in our response to reviewer 2’s final major point (above) with an analysis showing that lever-press latency was not significantly related to the calcium signals.

- In the Discussion, the authors should speculate on how unilateral dopamine neuron signals affect the contralateral side of the body (e.g., limbs or else) in order to initiate/support/perform a movement. This is a central part of the conclusion, if I am not mistaken, and should be honored with a speculation on how this may be implemented in terms of functional neuroanatomy. Also, rotation behavior after 6-OHDA lesion should be addressed in this context.

We appreciate your correctly intuiting our speculation and pointing out that it was not made explicit in the previous manuscript. We indeed envision that DA signals in each side might be important for initiating contralateral movement directly. This fits well with the classic picture of the functional anatomy of the basal ganglia (i.e., the direct and indirect pathways and their modulation by dopamine; (DeLong, 1990), together with the contralateral organization of the motor system, including striatum (Tai et al., 2012; Kitama et al., 1991). As the reviewer points out, there is also causal evidence for such a function: previous work has shown that unilateral excitation of DA neurons or neurons innervated by DA neurons has led to increased contralateral rotations or contralateral movement (Saunders et al., 2018). Moreover, classic results on unilateral lesions via 6-OHDA show that impairing DA neurons in one hemisphere of the brain led to increased ipsilateral rotations, further showing that the causal relationship between unilateral signals and contralateral movements (Costall, Naylor, and Pycock, 1976; Ungerstedt and Arbuthnott, 1970). We have included discussion of these points in the revised manuscript.

- In the Materials and methods section, it is stated that 1-5 recordings were obtained per recording site. Does that mean that some animals contributed a lot more data than others? For example, 10.108 "terminal" trials were recorded. That makes about 840 per animal on average. Is that roughly the average number of trials per animal? If not, it should be reported.

Thank you for this comment. We recorded on average 791.89 ± 371.80 (mean ± SD) trials per mouse. For VTA/SN::DMS Terminals recordings specifically, we had 842.33 ± 356.72 trials per mouse. For VTA/SN::DMS Cell-Bodies recordings specifically, we had 705.43 ± 381.10 trials per mouse. We have now included this information in the Materials and methods section.